# Blood Meal Sources of *Anopheles* spp. in Malaria Endemic Areas of Honduras

**DOI:** 10.3390/insects11070450

**Published:** 2020-07-16

**Authors:** Denis Escobar, Krisnaya Ascencio, Andrés Ortiz, Adalid Palma, Ana Sánchez, Gustavo Fontecha

**Affiliations:** 1Microbiology Research Institute, Universidad Nacional Autónoma de Honduras, Tegucigalpa 11101, Honduras; denis.escobar@unah.edu.hn (D.E.); kris_ascencio1996@hotmail.com (K.A.); aortizm@unah.edu.hn (A.O.); adalid1996.31@hotmail.com (A.P.); 2Department of Health Sciences, Brock University, St. Catharines, ON L2V 5A2, Canada; asanchez@brocku.ca

**Keywords:** *Anopheles* spp., blood meal source, malaria, Honduras, *Plasmodium* spp.

## Abstract

Malaria remains a life-threatening disease in many tropical countries. Honduras has successfully reduced malaria transmission as different control methods have been applied, focusing mainly on indoor mosquitoes. The selective pressure exerted by the use of insecticides inside the households could modify the feeding behavior of the mosquitoes, forcing them to search for available animal hosts outside the houses. These animal hosts in the peridomicile could consequently become an important factor in maintaining vector populations in endemic areas. Herein, we investigated the blood meal sources and *Plasmodium* spp. infection on anophelines collected outdoors in endemic areas of Honduras. Individual PCR reactions with species-specific primers were used to detect five feeding sources on 181 visibly engorged mosquitoes. In addition, a subset of these mosquitoes was chosen for pathogen analysis by a nested PCR approach. Most mosquitoes fed on multiple hosts (2 to 4), and 24.9% of mosquitoes had fed on a single host, animal or human. Chicken and bovine were the most frequent blood meal sources (29.5% and 27.5%, respectively). The average human blood index (HBI) was 22.1%. None of the mosquitoes were found to be infected with *Plasmodium* spp. Our results show the opportunistic and zoophilic behavior of *Anopheles* mosquitoes in Honduras.

## 1. Introduction

Malaria is a human parasitic disease caused by four species of *Plasmodia*, and its main transmission mechanism is through mosquito vectors of the genus *Anopheles*. Malaria continues to be a great burden on the public health and economy of many tropical countries [1]. Along with other countries in Mesoamerica, Honduras has established the goal of eliminating malaria by 2030 [2], a goal that seems within reach, since the country has managed to substantially reduce vectorial transmission by more than 96% since 2004, reporting only 651 cases in 2018 [1]. The strategies implemented to achieve this goal have included the timely diagnosis and treatment of symptomatic cases, surveillance of active foci, pharmacovigilance, indoor residual spraying, and the use of long-lasting insecticide-treated mosquito nets in some regions. Measures aimed at vector control have played a major role. Integrated vector control interventions for *Anopheles* species are key components in achieving malaria elimination worldwide [3]. 

Twelve species of *Anopheles* have been officially registered by the Honduran health authorities in entomological surveillance reports. According to these surveys, as well as international databases [4], the most frequent and widely distributed species in the country is *Anopheles albimanus*. As a generalist species which requires no specific habitat, *An. albimanus* can be widely dispersed, and is considered a dominant vector in Central America [5,6,7]. This information has been confirmed by a recent publication describing seven *Anopheles* species from endemic areas in Honduras, where 74% of the mosquitoes were identified as *An. albimanus* [8]. Other secondary vectors were also reported, but with a more limited geographical distribution.

Differences in the distribution of dominant and secondary vector species, as well as their vector capacity, contribute substantially to malaria endemicity. Vector capacity (i.e., the ability of a vector to transmit a pathogen) is defined by the sum of several factors, such as anthropophilic behavior [9], intrinsic mortality [10], indoor resting [11], biting hours [12], length of the gonotrophic cycle, and the gonotrophic discordance [13]. A further key element determining vector capacity is the insect´s feeding behavior. Some species of mosquitoes, for example, are opportunistic, and can feed on multiple sources (human and animals), depending on hosts’ availability [14,15,16]. Feeding on mixed blood meals from different hosts within a single gonotrophic cycle is also a common behavior for some *Anopheles* species [17]. This could contribute to maintaining vector populations conducing to high vector densities, which could increase malaria transmission in a given geographic area.

The two main insecticide-based interventions for malaria elimination implemented in Honduras and many other countries are indoor residual spraying (IRS) and long-lasting insecticide treated nets (LLIN) [18]. Despite their effectiveness, not only do these interventions neglect potential vectors that feed and rest outdoors, but they can also contribute to a selective pressure that may translate into behavior change. It has been shown that, under indoor insecticide pressure, mosquitoes are forced to feed on animals in the peridomicile, adjusting their biting preference towards more available hosts [19]. The evaluation of the vector competence and identification of pathogen reservoirs requires a solid understanding of the vector’s population dynamics after insecticide-based interventions, as well as the preferences of vectors for specific hosts. In Honduras, no information is available regarding the food sources of *Anopheles* species. To contribute to this knowledge, the present study aimed to investigate the blood meal sources of anophelines resting outdoors, as well as the proportion of infected mosquitoes in endemic areas of Honduras.

## 2. Materials and Methods

### 2.1. Mosquito Collection

Anopheline mosquitoes were collected between February and October 2019, at eight sites in five departments in Honduras. Capture sites were located near small rural villages, in which agricultural and fishing activities take place. Six out of the eight collection sites were located in three departments near the Caribbean region with very humid tropical climates (Atlántida, Colón and Gracias a Dios), while the other two sites were located in the dry central tropical region (Comayagua, El Paraíso) (Figure 1, Appendix A). Of the six collection sites in the Caribbean region, two were located in Gracias a Dios, a department commonly known as La Mosquitia, which is an ecological region geographically isolated from the rest of the country by the Rio Plátano biosphere. Before the entomological collection, the species of domestic animals present at each site within a radius of 200 m from the collection sites were recorded. Major economic activities at the eight sites were also recorded.

Entomological collections were carried out using two methods. The first method used CDC light traps with no other attractant but common light. Three to five traps were placed per site in the outdoor structures of human dwellings, as well as in structures for domestic animals’ rest. CDC light traps were set for 12 h between 6:00 p.m. and 6:00 a.m. Traps were separated a minimum of 50 m from each other. The second method consisted of manual aspiration of the anophelines in outdoor areas were animals were resting and outside the households, between 6:00 p.m. and 9:00 p.m. Mosquitoes at rest on surfaces were aspirated using mouth aspirators with HEPA filters, model 612 (John W. Hock Company). Each mosquito collection was conducted for one night at each site [20]. After each collection, the insects captured by either method were placed in plastic bags and frozen at −20 °C. The following days, anophelines were separated and placed in Petri dishes with silica gel, and then transported at room temperature to the laboratory, where they were stored at −20 °C until morphological identification.

### 2.2. Identification of Mosquito Species

The morphological identification of mosquito species was done using taxonomic keys for anophelines of Central America, according to standard procedures [21]. Blood feeding status was visually assessed on each mosquito by the appearance of the abdomen. Wings and legs were preserved as vouchers at the entomological collection of the National University of Honduras (UNAH). Mosquitoes were individually stored at −20 °C for further molecular tests.

### 2.3. DNA Extraction and Blood Meal Identification

Engorged mosquitoes of seven species from all collection sites were chosen for blood meal analysis. DNA was extracted with the AxyPrep MAG Tissue-Blood gDNA Kit, Axygen^®^ (Corning Incorporated, Life Sciences, Tewksbury, MA, USA). Immediately before DNA extraction, mosquitoes were individually macerated with a pestle in a 1.5 mL tube, with 50 µL of lysis solution included in the kit. Molecular tests were carried out for each mosquito, in order to detect five possible food sources. Individual PCR reactions were carried out for each animal blood source. Species-specific primers were used, as described by Pizarro et al., to amplify short interspersed nuclear elements (SINEs) of *Sus scrufa* (pig), *Gallus gallus* (chicken), *Bos taurus* (bovine) and *Canis familiaris* (dog) [22]. The detection of human DNA was carried out using a region of the beta-globin gene [23]. Each experiment was performed using positive and negative controls. Details of PCR conditions and components are described on Table 1. 

PCR amplifications for bovine, pig, chicken and dog were carried out in a volume of 20 µL with 10 µL of Taq Master Mix 2X (Promega, Madison, Wisconsin), 0.8 µL of each primer (10 µM), 1–2 µL of DNA, and nuclease-free water. PCR reactions for human DNA were performed in a volume of 20 µL with 10 µL of Taq Master Mix 2X, 2 µL of each primer (10 µM), 2 µL of DNA, and nuclease-free water. 

PCR cycling conditions for the amplification of DNA from non-human sources were as follows: one cycle at 95 °C for 10 min, followed by 35 cycles at 95 °C for 30 s, annealing temperatures as shown in Table 1 for 30 s, 72 °C for 30 s, and a final cycle at 72 °C for 7 min. Human DNA amplifications were as follows: one cycle at 94 °C for 5 min, 40 cycles at 94 °C for 1 min, 60 °C for 1 min, 72 °C for 1 min, and one cycle at 72 °C for 10 min. Amplification products were separated and visualized by electrophoresis in 2% agarose gel with ethidium bromide.

The number of each anopheline species and their corresponding blood meal source was recorded. In addition, the human blood index (HBI) was calculated as the crude mean proportion of individuals of each species found to contain human blood—including those with multiple blood meal sources—divided by the total number of positive blood-fed mosquitoes [24,25]. The number of mosquitoes containing blood from a single host was also recorded. Mosquitoes that had fed from two or more hosts were classified as mixed blood meals.

### 2.4. Quantitative Interaction Network

A quantitative interaction network plot and an interaction matrix plot were constructed to generate graphic images, showing relationships between *Anopheles* species and host preferences. Both graphics were constructed using *plotweb* and *viswed* functions of Bipartite package of R, with default parameters [26].

### 2.5. Detection of Plasmodium spp. DNA

In order to detect the parasite’s DNA in engorged mosquitoes, a subset of anophelines from Gracias a Dios was chosen randomly. This department was selected because it is the main hotspot of malaria transmission in the country. In total, 36 specimens were analyzed: 25 *Anopheles albimanus* and 11 *An. crucians*. 

Detection of malaria parasites was based on amplification of the 18s rRNA gene of the genus *Plasmodium* spp. through a nested PCR approach as described in previous studies [27]. Briefly, the first PCR was carried out in a volume of 25 µL, with 12.5 µL of Taq Master Mix 2X, 1 µL of primers rPLU1 and rPLU5 (10 µM), 5 µL of DNA, and nuclease-free water. A second PCR used 1 µL of primers rPLU3 and rPLU4 (10 µM), was performed under the same conditions as above, but using 1 µL of DNA from the first reaction as a template. 

Both PCR reactions were carried out as follows: one cycle at 94 °C for 4 min, followed by 35 cycles at 94 °C for 30 s, 55 °C (PCR 1) and 62 °C (PCR 2) for 1 min, 72 °C for 1 min, and a final extension at 72 °C for 4 min.

Positive and negative DNA controls of *Plasmodium vivax* and *P. falciparum* were included within each experiment. Amplification products were separated and visualized by electrophoresis in 1% agarose gels with ethidium bromide.

## 3. Results

### 3.1. Description of the Collection Sites

Bovines, dogs, pigs and chickens were observed around households at all collection sites. All departments, except for Gracias a Dios, have significant agricultural and livestock production, with rice, sugar cane and banana as the main crops. Subsistence animal husbandry is common, with many families owning chickens and pigs, and cattle to a lesser extent. On the other hand, the main economic activity of Gracias a Dios inhabitants is fishing.

### 3.2. Blood Meal Identification

Overall, 311 anopheline females were collected. Of these, only 181 (58.2%) female mosquitoes of seven *Anopheles* species were visible engorged and separated for analysis (Table 2). Overall, 130 (41.8%) female mosquitoes were not visibly blood-fed. The DNA of five potential vertebrate hosts was amplified for all the engorged mosquitoes. The most frequent blood meal sources were *Gallus gallus* (chicken) (29.5%) and *Bos taurus* (bovine) (27.5%). *Canis familiaris* (dog) was the least preferred host (11.9%). Forty anophelines of four species were positive for human blood, with an average HBI of 22.1%. The highest HBI was found in *Anopheles darlingi* (55%), followed by *An. albimanus* and *An. pseudopunctipennis*, with an HBI of 25% each. *An. vestitipennis* showed an HBI of 4.4%. In three species (*An. crucians*, *An. neivai* and *An. punctimacula*), the presence of human blood could not be demonstrated. 

*Anopheles albimanus* showed the highest range of blood meal sources (n = 5) (Figure 2). In *An. darlingi* and *An. vestitipennis* four hosts were demonstrated. Three different hosts were detected for *An. crucians*. Two blood meal sources were detected in *An. neivai*, while in *An. punctimacula*, only bovine blood was found. Human blood was detected in one specimen of *An. pseudopunctipennis*.

Figure 3 shows a network and matrix of interactions between the anopheline species and their blood meal sources. 

The diversity of hosts was also analyzed according to geographic location (Figure 4). There were some differences in the proportion of blood sources between four species of anophelines with more than 20 individuals. Three *Anopheles* species with less than 20 individuals were not analyzed due to the low number of specimens. *Anopheles albimanus* mosquitoes from five collection sites were analyzed. The proportion of mosquitoes that fed on chicken was higher in Comayagua, whereas the most frequent blood meal source in El Paraíso was pig. Specimens of *An. darlingi* were collected only in Atlántida and Colón. In both localities, the most frequent blood meal sources were bovine and chicken. 

In addition, 29.8% of the blood meals were of unidentified origin (Table 2, Figure 2). This means that 54 of 181 visibly engorged females did not amplify for any of the five vertebrate hosts tested. The largest number of cryptic meals was observed in Gracias a Dios (n = 42) for *An. albimanus*, *An. crucians* and *An. vestitipennis*. For the latter two species, the blood source of most individuals could not be identified (Figure 4). 

### 3.3. Number of Host Blood Meals

A total of 181 blood-fed *Anopheles* mosquitoes were analyzed for blood meal sources. Data summarized in Table 3 show the number and percentages of single-host blood meals and mixed blood meals from seven anopheline species. Most mosquitoes had fed on more than one host. The percentage of anophelines fed from a single animal source was 24.9%, while 27.6% fed on two different animal sources. Mosquitoes fed from three and four different animals were also detected in a smaller proportion. It is remarkable that 40% of the females of *An. darlingi* fed on four different blood sources. Only six specimens showed human blood as the only meal source: four *An. albimanus*, one *An. vestitipennis* and one *An. pseudopunctipennis*.

Further analyses were done to determine if females without visible engorgement had also ingested blood from a vertebrate host. Thus, 58 mosquitoes of seven species were randomly selected and analyzed, resulting in 10.3% (6/58) amplifying for at least one host. Three mosquitoes (5.2%) had fed on bovine, while three others had fed on pig, chicken or dog. No human DNA was detected in any of these mosquitoes.

### 3.4. Parasite DNA Detection

None of the 36 *Anopheles* mosquitoes analyzed tested positive for *Plasmodium* spp. DNA.

## 4. Discussion

The identification of mosquito feeding preferences is important in order to understand the relevance of non-human blood-meal sources on maintaining vector populations. It also helps to evaluate changes in mosquito behavior in response to indoor interventions. Surveillance protocols and mosquito control interventions focus primarily on endophilic and endophagic vectors, underestimating the influence of those that feed and rest outdoors. It has been suggested that hosts in peridomiciliary areas could contribute significantly to maintaining high densities of mosquito populations, and consequently could contribute to malaria transmission, especially in countries where much effort has been invested in indoor vector control [16,28]. 

In this study, mosquitoes were collected outside households, and under the structures where domestic animals spend the night. This approach, although limited—as it did not include mosquitoes resting indoors—was adequate to meet our goal of collecting mosquitoes that had fed on animals in the peridomicile, as well as those that had fed on humans and were resting outdoors. A large majority of blood meals (204/244) were of animal blood, and less than 17% (40/244) were of human origin. The average human blood index (HBI) was 22.1%. These findings are similar to those previously published in Mexico, where the host selection patterns of *An. albimanus* collected indoors and outdoors were analyzed [29]. Due to the design of our study, it is not possible to elucidate whether mosquitoes that had fed on human blood did so indoors and then went outside to rest, or if they had actually fed on humans outdoors. In this study, human blood was not found in *An. crucians*, *An. neivai* or *An. punctimacula*, species considered as secondary vectors and predominantly zoophilic [30,31]. Since mosquitoes were caught outdoors at night, the probability of having sourced their blood meal from humans is lower in relation to the number of animals available in the peridomicile. Therefore, a greater affinity for animals as blood source could be explained by the availability of hosts, collection frequency, trapping method and trapping location [32]. Given the low number of specimens captured from the three species, these hypotheses need to be confirmed later with a larger number.

Some *Anopheles* species have shown a strong preference for humans as blood source, particularly in Africa [33,34], while other species display higher blood-host plasticity [15,35]. Not all species have a high preference for human blood. As shown by Massebo et al., some species exhibited zoophagic behavior, despite the large human populations available compared to that of domestic animals [36]. In contrast, several studies have demonstrated that some anophelines select their hosts depending on their availability, and not due to strict species tropism. Orsborne et al. conducted a systematic review and meta-regression of three major malaria vectors in Africa, and showed that HBI was more associated with location of mosquito captures than with mosquito species [19]. A study conducted in Cameroon concluded that when *Anopheles rufipes* finds alternative hosts to feed, its anthropophagic behavior decreases [37]. Another study involving *Anopheles stephensi* in India found that resting mosquitoes were more prevalent in cattle sheds than in human houses. Here, the authors propose that the determining factor of this behavior was the easy availability of blood meal sources [38].

Our results show that chicken and cattle were the most frequent blood meal sources. This is not surprising, since these are the most commonly found animals in the country’s rural households of Honduras, with the exception of La Mosquitia, where cattle are unusual. Our findings indicate that the anopheline species analyzed did not show a well-defined preference for any particular host. These data seem to support the premise that blood meal intake reflects host availability rather than host preference [39,40], and provide evidence that these vectors tend to be exophagic, exophylic and zoophagic [41].

The more frequent hosts detected in *An. darlingi* were chicken and cattle, followed by humans and dogs. This result differs from what has been reported for this species in South America, where a more anthropophilic behavior appears to be the most common, albeit combined with opportunistic zoophilic feeding [14,16,25,42]. Differences (genetic and/or behavioral) between *An. darlingi* populations from Mesoamerica and South America could be explained by geographic isolation [8]. On the other hand, the results we obtained for *An. albimanus* are in agreement with previous reports from other Latin American countries, which highlight zoophagic and opportunistic preferences [24,29]. Unfortunately, studies on blood meal sources of malaria vectors in the Americas are outdated, scarce, or simply non-existent. Thus, some of the results presented here are not readily comparable with the literature. This highlights a knowledge gap in this field that should be addressed.

Most of the mosquitoes proved to be feeding on more than one host (2 to 4), and only 24.9% of mosquitoes were feeding on a single host species, animal or human. Many reports show similar behavior of feeding on multiple human hosts [43,44], or different animal species in Africa [15,37,45,46,47], Asia [48], Oceania [44], and Latin America [9,14,16,25,29]. Reports of blood meals of single-host origin in engorged anophelines are less frequent [49]. Consequently, our results support the phenomenon of gonotrophic discordance [13], indicating that two or more successive blood meals are common within a single gonotrophic cycle for *Anopheles* species in Honduras; a potential reproductive strategy to increase fecundity [50].

Out of a total of 181 visibly engorged mosquitoes, less than half (n = 54, 29.8%) did not amplify for any of the five animal blood sources analyzed. We can offer two possible explanations for this result. Firstly, host DNA could have been degraded by rapid digestion [51]. This explanation is unlikely, given that mosquitoes were collected and killed only a few hours after feeding, and the DNA of the hosts appears to be stable for a long time [52,53]. A second and more plausible explanation is the existence of other available animal hosts from which mosquitoes obtain their blood meals. This phenomenon has been widely described, both in the Amazon basin [4] and Africa [35,54,55]. It is worth noticing that a majority of unidentified blood meal sources was observed in specimens from La Mosquitia (Gracias a Dios), specifically in *An. crucians* and *An. vestitipennis*. Since this geographical region is a protected biosphere with little human intervention, there exist a highly diverse wild fauna accessible to mosquito bites. Testing for only five blood meal sources is a limitation to our study. Future investigations should consider the use of generic primers from mammals or other groups of animals and subsequent sequencing, to discover unusual wild hosts.

A strength of the present study lies on the analysis of 58 female mosquitoes without visible engorgement. A recent publication investigated the presence of host DNA in 217 visibly unfed *Anopheles* mosquitoes from Madagascar, and the authors found that 74% had fed on a mammal [56]. Since almost all studies on vector meal preferences utilize visually engorged mosquitoes, it is reasonable to expect an underestimation of the proportion of host sources. Of the seven *Anopheles* species we analyzed, 10.3% were positive for at least one host (bovine, pig, chicken or dog). We recommend that future studies do not preclude visibly unfed mosquitoes from blood meal analysis.

Finally, we were unable to detect *Plasmodium* DNA in the analyzed specimens. Several studies have been successful to detect *Plasmodium* spp. antigens in the engorged females of many anopheline species in highly endemic settings [14,34,37,38,57]. In this study, 36 mosquitoes were tested for *Plasmodium* DNA, but none were positive. This is likely due to the low incidence of malaria cases in the studied communities [1] (Appendix A). To increase sensitivity in *Plasmodium* detection, further studies should increase the number of mosquito specimens, as well as include vectors found inside the households (i.e., endophilic and endophagic vectors).

## 5. Conclusions

An analysis of the blood meals of mosquitoes resting outdoors revealed that all the anopheline species feed mainly on domestic animals commonly found in the country: chicken, pigs, bovines, and dogs. The exception of this finding was in La Mosquitia, where the main food sources remain unidentified. Two or more successive blood meals were also common within each gonotrophic cycle, a behavior strategy that could increase insect fecundity. There does not seem to be a clear preference for any host among the anopheline species analyzed. These results support the hypothesis that malaria vectors in Honduras exhibit opportunistic feeding behavior and underscore the need for additional mosquito-control measures focusing on the peridomestic environment. Such integrated approach would yield more comprehensive data to inform malaria elimination efforts in the country.

## Figures and Tables

**Figure 1 insects-11-00450-f001:**
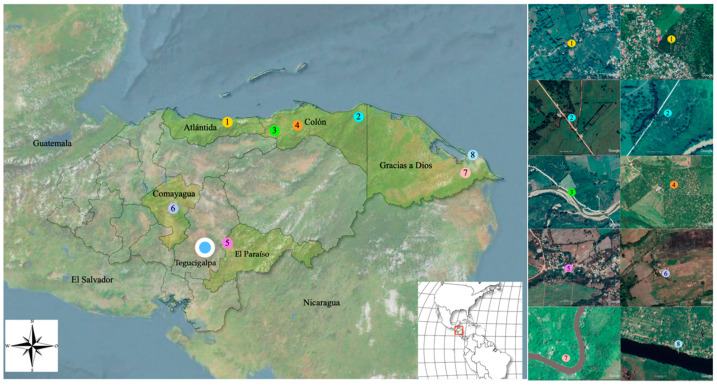
Map of the sites where the mosquito collections were performed in five departments of Honduras. (1) La Ceiba; (2) Iriona; (3) Sonaguera; (4) Tocoa; (5) El Paraíso; (6) Comayagua; (7) Tikirraya; (8) Kaukira. Images from Google Earth.

**Figure 2 insects-11-00450-f002:**
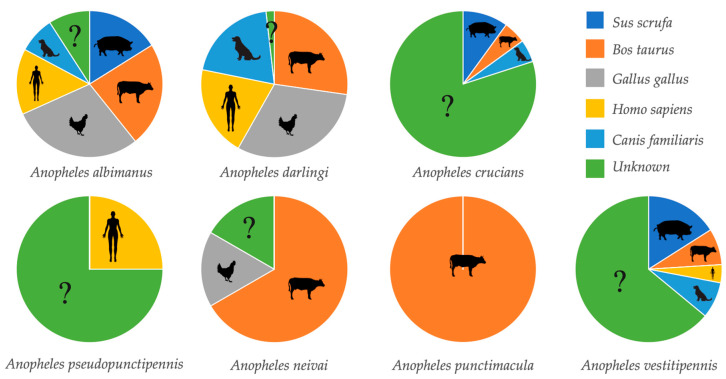
Proportion of blood meal sources by anopheline species.

**Figure 3 insects-11-00450-f003:**
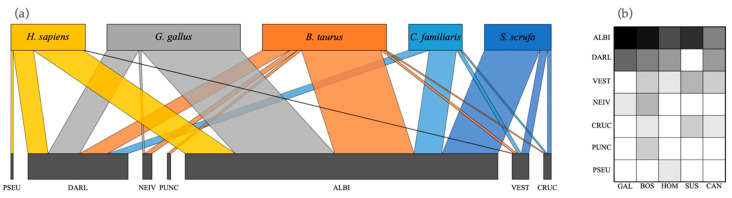
(**a**) Quantitative interaction networks, and (**b**) interaction matrix of blood-meal sources for seven *Anopheles* species. Network is based on the analysis of blood-meal source for 181 specimens. PSEU = *An. pseudopunctipennis*; DARL = *An. darlingi*; NEIV = *An. neivai*; PUNC = *An. punctimacula*; ALBI = *An. albimanus*; VEST = *An. vestitipennis*; CRUC = *An. crucians*.

**Figure 4 insects-11-00450-f004:**
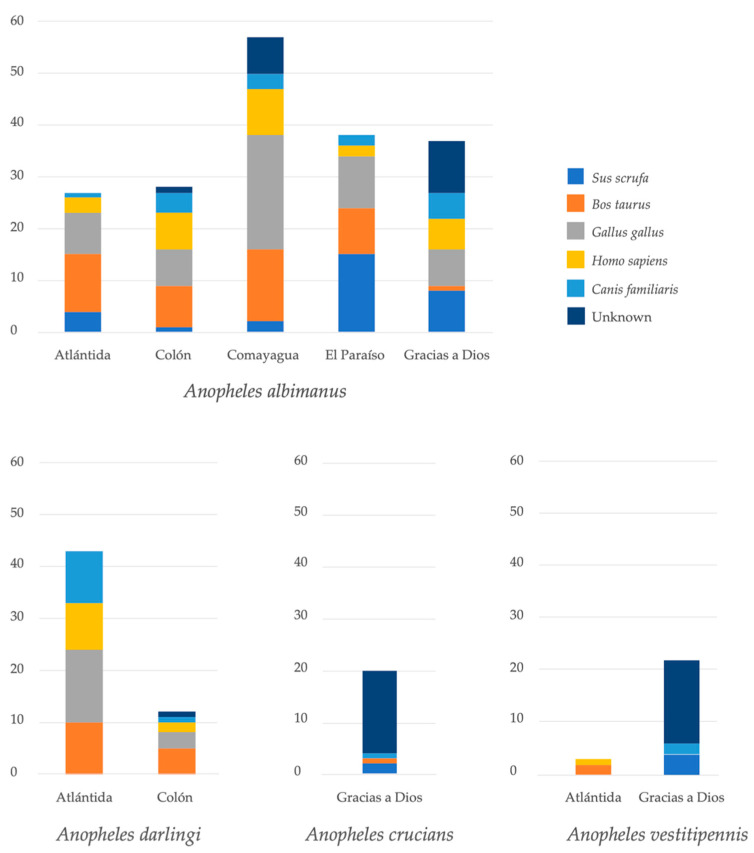
Source of blood meal of four *Anopheles* spp. according to collection site.

**Table 1 insects-11-00450-t001:** Primers, annealing temperature and product sizes of PCR reactions.

Host	Forward Primer Sequence (5′-3′)	Reverse Primer Sequence (5′-3′)	Annealing T° (°C)	Product Size (bp)
Human	acacaactgtgtttcactagc	gaaacccaagagtcttctct	60	210
Dog	agggcgcgatcctggagac	agacacaggcagagggagaa	58	83
Bovine	tttcttgttatagcccaccacac	tttctctaaaggtggttggtcag	60	98
Chicken	ctgggttgaaaaggaccacagt	gtgacgcactgaacaggttg	58	169
Pig	gactaggaaccatgaggttgcg	agcctacaccacagccacag	60	134

**Table 2 insects-11-00450-t002:** Blood meal origins and Human Blood Index (HBI) in seven *Anopheles* species.

	*Sus scrufa*	*Bos taurus*	*Gallus gallus*	*Homo sapiens*	*Canis familiaris*	HBI (%)
*An. albimanus*	30	43	54	27	15	25.2
*An. darlingi*	-	15	17	11	11	55
*An. crucians*	2	1	-	-	1	-
*An. neivai*	-	4	1	-	-	-
*An. pseudopunctipennis*	-	-	-	1	-	25
*An. punctimacula*	-	2	-	-	-	-
*An. vestitipennis*	4	2	-	1	2	4.4
Total	36	67	72	40	29	22.1

**Table 3 insects-11-00450-t003:** Number of single-host blood meals, mixed blood meals and blood meals of unknown origin from seven anopheline species.

	n	Single (n = 1)	%	Mixed (n = 2)	%	Mixed (n = 3)	%	Mixed (n = 4)	%	Unknown	%
*An. albimanus*	107	27	25.2	42	39.3	20	18.7	1	0.9	17	15.9
*An. darlingi*	20	3	15.0	5	25.0	3	15.0	8	40.0	1	5.0
*An. crucians*	20	4	20.0	0	0.0	0	0.0	0	0.0	16	80.0
*An. neivai*	5	3	60.0	1	20.0	0	0.0	0	0.0	1	20.0
*An. pseudopunctipennis*	4	1	25.0	0	0.0	0	0.0	0	0.0	3	75.0
*An. punctimacula*	2	2	100.0	0	0.0	0	0.0	0	0.0	0	0.0
*An. vestitipennis*	23	5	21.7	2	8.7	0	0.0	0	0.0	16	69.6
Total	181	45		50		23		9		54

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
