# Peer review of "Blood Meal Sources of *Anopheles* spp. in Malaria Endemic Areas of Honduras"

_insects, 2020, doi:10.3390/insects11070450_

Round 1
Reviewer 1 Report
the manuscript is interesting and brings information that can be used to help the development of new strategies for malaria control.
In the introduction, there is almost no information about Honduras malaria situation. would be great to see the drop of cases over the years and the methods used to control the disease.
The non-engorged female issue, if there is no method problem, should be better explored in the manuscript. I feel that it was not explored.
Overall I think that the location should place a critical point (as stated for the Gracias a Dios), and the analysis didn't considered this information.
The collections frequency is also not clear, and I would expect to see a difference on population during different seasons (or something proving there is no difference). I suppose that for each site the collection happened each month (Feb. to Oct. 2019), but they were not collected simultaneously.
The second attached document, contains a lot of information that would be quite interesting on the main manuscript and providing a substantial uniqueness of it, please reconsidered including part of that material in the main manuscript.
I am sending the PDF with my comments and suggestions.

Author Response
Point-by-point responses:
Reviewer 1
- In the introduction, there is almost no information about Honduras malaria situation. would be great to see the drop of cases over the years and the methods used to control the disease.
- Answer. The first paragraph of the introduction has been expanded with the following information: “Malaria is a human parasitic disease caused by four species of Plasmodia and its main transmission mechanism is through mosquito vectors of the genus Anopheles. Malaria continues to be a great burden on the public health and economy of many tropical countries [1]. Along with other countries in Mesoamerica, Honduras has established the goal of eliminating malaria by 2030 [2]; a goal that seems within reach, as the country has managed to substantially reduce vectorial transmission by more than 96% since 2004, reporting only 651 cases in 2018 [1]. The strategies implemented to achieve this goal have included timely diagnosis and treatment of symptomatic cases, surveillance of active foci, pharmacovigilance, indoor residual spraying, and the use of mosquito nets in some regions. Measures aimed at vector control have played a major role. Integrated vector control interventions for Anopheles species are key components in achieving malaria elimination worldwide [3].
- The non-engorged female issue, if there is no method problem, should be better explored in the manuscript. I feel that it was not explored.
- Answer. We are confident that there were no method problems. Each experiment was validated by positive and negative controls. However, the reviewer is correct. This finding is extremely interesting, and deserves a further approach; however, we did not want to explore this further because we were at risk of clouding the results of the main research question of this study. Future research will aim to further explore the issue of non-engorged females.
- Overall I think that the location should place a critical point (as stated for the Gracias a Dios), and the analysis didn't considered this information.
- Answer. The analysis did not consider the issue of the location in depth because the objective of our study was not to establish a correlation between the ecological conditions of the capture sites and the intake of the anophelines. In the case of Gracias a Dios a special mention was made because it is an area with completely different social, cultural and ecological characteristics from the rest of the country.
- The collections frequency is also not clear, and I would expect to see a difference on population during different seasons (or something proving there is no difference). I suppose that for each site the collection happened each month (Feb. to Oct. 2019), but they were not collected simultaneously.
- Answer. The following information has been added to the section 2.1: “The second method consisted of manual aspiration of the anophelines in outdoor areas were animals were resting and outside the households, between 6:00 pm and 9:00 pm. Mosquitoes at rest on surfaces were aspirated using mouth aspirators with HEPA filter, model 612 (John W. Hock Company). Each collection was carried out for one night per site [20]. After each collection, the insects captured by either method were placed in plastic bags and frozen at -20 °C. The following days, anophelines were separated and placed in Petri dishes with silica gel and then transported at room temperature to the laboratory where they were stored at -20 °C until morphological identification.” The reviewer is correct. Insect captures were not performed simultaneously, therefore differences between populations of anopheline species would be expected; however, although it is a very interesting research question, the comparison of the fluctuations of the species between seasons along the year was not an objective of this study. This study was intended to describe the blood intake of the Anopheles species found at each site and collected at different time intervals.
- The second attached document, contains a lot of information that would be quite interesting on the main manuscript and providing a substantial uniqueness of it, please reconsidered including part of that material in the main manuscript.
- Answer. If the reviewer refers to the paper that was uploaded as a proof of the “unpublished data” reference and still in the process of being published, that document was already published on July 1, 2020 in Parasites & Vectors, and updated in the manuscript.
- Line 38. Is there a reference or public access material, or any other reference type to include here? Or this work was conducted by the group? In the last case, should be mentioned.
- Answer. This information has been collected by technicians from the country's Ministry of Health (not by the authors), which publishes them for internal purposes as part of routine entomological surveillance since 2013. According to those reports, 12 species of anophelines have been identified through morphometric keys. However, this information is not citable because it is not available to the public.
- Lines 41-43 If there is a publication, it should be cited. If the study was not yet published, and it was not performed by the same group, then this information should be removed, once there is no way to confirm. Is this information also stated from the previous study? Please provide a reference for this statement or remove the sentence.
- Answer. The reference has been updated with the recent publication in Parasites and Vectors, Escobar et al 2020.
- Lines 46-48 Please consider to merge and maybe also invert the position of the sentence. Right now it is disconnected and in an unfriendly order.
- Answer. The sentences have been modified as follows in a more “friendly” redaction: Differences in the distribution of dominant and secondary vector species, as well as their vector capacity, contribute substantially to malaria endemicity. Vector capacity (i.e., the ability of a vector to transmit a pathogen) is defined by the sum of several factors, such as anthropophilic behavior [9], intrinsic mortality [10], indoors resting [11], biting hours [12], length of the gonotrophic cycle, and the gonotrophic discordance [13].
- Lines 52-53 Isn't it true that not all plasmodium can infect different hosts? For example, the P. gallinaceum cannot develop in humans and some can be found in humans and monkeys (P. knowlesi). So, what you are saying is that having a bigger range of options to feed, it would increase the parasite transmission rate? Is there any modeling study considering this aspect?
- Answer. What we would like to convey is that the availability of hosts (on which anophelines can be fed) could contribute to maintaining vector populations. High populations densities of vectors favor the maintenance of malaria transmission in a given geographic area. The paragraph has been clarified as follows: “Mixed blood meals from different hosts within a single gonotrophic cycle is also a common behavior for some Anopheles species [17]. This could contribute to maintaining vector populations conducing to high vector densities, which could increase malaria transmission in a given geographic area.”
- Lines 55. Are these two the main interventions in Honduras?
- Answer. The sentence has been modified as follows: “The two main insecticide-based interventions for malaria elimination in Honduras and many other countries are indoor residual spraying (IRS) and long-lasting insecticide treated nets (LLIN) [18]”.
- Line 57-58. Can also contribute to a selective pressure that reflects a change in the behavior.
- Answer. The sentence has been changed as kindly suggested by the reviewer.
- Line 75 200 m radius from exactly where?
- Answer. Sentence has been modified: “Before the entomological collection, the species of domestic animals present at each site within a radius of 200 m from the collection site were recorded.”
- Line 77 The CDC light trap can use both, UV and (originally) common light. In this case, which light was used?
- Answer. Sentence modified as follows: “The first method used CDC Light traps with no other attractant but common light.”
- Line 78 Were they scattered around the same place? what was the area coverage? what were the criteria to place the traps? what was the frequency of trapping (the study was from Feb. to Oct. 2019)?
- Answer. The reviewer´s question stems from our lack of clarity in that section of the manuscript. A single capture was made at each selected site. Therefore, a frequency cannot be defined in the trapping. For clarity, the following two sentences have been included in section 2.1: “Capture sites were located near small rural villages in which agricultural and fishing activities take place.”; and “Traps were separated a minimum of 50 meters from each other.”
- Line 80 how many people were doing the aspiration (number of aspirators) were these aspirators next to the CDC traps? how this was avoided?
- Answer. Three to four people were in charge of the aspiration. Each person had their own aspirator. Aspirations were performed in the same areas where the CDC traps were placed.
- Line 82 How this was done? you would need to knock them down and to keep them chilled, in order to remove only the anophelines.
- Answer. The insects were placed in bags and frozen at -20 ºC immediately after being collected. The separation of anophelines from the rest of the insects was carried out during the subsequent days or weeks after being transported to the Tegucigalpa laboratory.
- Line 82 This was regardless of the trapping/collecting method? considering only CDC trap, we would have one night per month and per location and minimum three traps each site, resulting in 192 samples, and not all samples were positive for Anopheles and for engorged female. Is this conclusion corrected? It would be nice to have a better description of the collection method.
- Answer. This paragraph has been expanded as follows: “Anopheline mosquitoes were collected between February and October 2019 at eight sites in five departments in Honduras. Capture sites were located near small rural villages in which agricultural and fishing activities take place. Six out of the eight collection sites were located in three departments near the Caribbean region with very humid tropical climate (Atlántida, Colón and Gracias a Dios), while the other two sites were located in the dry central tropical region (Comayagua, El Paraíso) (Fig. 1, supplementary table 1). Of the six collection sites in the Caribbean region, two were located in Gracias a Dios, a department commonly known as La Mosquitia, which is an ecological region geographically isolated from the rest of the country by the Rio Plátano biosphere. Before the entomological collection, the species of domestic animals present at each site within a radius of 200 m from the collection sites were recorded. Temperature, relative humidity, and major economic activities at the eight sites were also recorded. Entomological collections were carried out using two methods. The first method used CDC Light traps with no other attractant but common light. Three to five traps were placed per site in the outdoor structures of human dwellings as well as in structures for domestic animals’ rest. CDC light traps were set for 12 hours between 6:00 pm and 6:00 am. Traps were separated a minimum of 50 meters from each other. The second method consisted of manual aspiration of the anophelines in outdoor areas were animals were resting and outside the households, between 6:00 pm and 9:00 pm. Mosquitoes at rest on surfaces were aspirated using mouth aspirators with HEPA filter, model 612 (John W. Hock Company). Each collection was carried out for one night per site [20]. After each collection, the insects captured by either method were placed in plastic bags and frozen at -20 °C. The following days, anophelines were separated and placed in Petri dishes with silica gel and then transported at room temperature to the laboratory where they were stored at -20 °C until morphological identification.”
- Line 86 It would be nice to have more information about these areas, how densely populated are they, number of malaria cases, distance from urban areas and the geographical position of each of them
- Answer. Capture sites were located near small rural villages (sparsely populated). A supplementary table has been included indicating location and coordinates of all capture sites and the number of malaria cases during 2018.
- Line 91 Was there any criteria to determine the feeding status? what was done with apparent unfed females?
- Answer. Feeding status was visually determined by the dilated state of the abdomen. The apparent unfed females were processed in the same way as the fed individuals.
- Line 110 Is this difference caused by the expected amplification size?
- Answer. The differences in the procedures correspond only to the availability of primers in our laboratory, and to the prior optimization for each procedure.
- Line 131 Why only a subset?
- Answer. Only one subset was chosen due to time constraints. Our collaborators were performing these experiments when our laboratory was closed due to the covid-19 pandemic. Given that the laboratory may be closed the rest of the year, we thought it convenient to publish the data obtained to date.
- Line 132 Please provide the information and threshold to select only mosquitoes from this area
- Answer. As indicated before, only mosquitoes from GaD were selected because it is the department in which the highest number of malaria cases is reported, and this increased the possibilities of demonstrating the parasite in the vector. Originally, the authors had set out to analyze a larger number of individuals, which was interrupted by the current pandemic. Consequently, the mosquitoes analyzed were randomly selected.
- Line 150 Have you seen an increase on other blood type that can be related to this observation.
- Answer. We do not understand this observation, Could the reviewer better explain his/her question?
- Lin 163 Can this be related to the capture method? would be very interesting to know the proportion of engorged females for each method.
- Answer. Indeed. It would have been interesting, but we weren't able to see it during the mosquito processing and unfortunately, we do not have that information.
- Lin 172 This table is quite confusing, because it shows the total amounts, but some mosquitoes had more than one blood type.
- Answer. This is true. The table can be a bit confusing, however the information it contains is understandable if you analyze it carefully. “n” represents the number of individuals analyzed for each species, but since the intake of multiple hosts was a very common phenomenon, the sum of the intakes does not correspond to the number of mosquitoes. Anyhow, the column expressing “N” has been eliminated.
- Line 178 There is no need for the legend with the figures representing each slice. in this case would be more interesting to have the species names instead of the letters
- Answer. Figure has been changed as suggested.
- Line 184 Include the short name of the species in the legend. Would be very interesting to see similar figures by location.
- Answer. The names of the species have been included in the legend. We agree; it would be interesting to build other similar figures, but that information is already contained in figure 4.
- Line 200 this figure should bring the species name for each graph, there is a lot of space
- Answer. Done.
- Line 221 This can only be true if the identification of alternative blood source is close to human. no primates were investigated in the study.
- Answer. As mentioned before, this idea is not intended to indicate that other animal hosts are capable of becoming infected with human plasmodia species, but rather that the availability of alternate blood sources contributes to maintaining high population densities of the vectors. These high mosquito densities are an important factor for the transmission of malaria in humans.
- Line 229 the study also do not differentiate these two locations in the analysis, or states that there was no difference
- Answer. This is correct. This study was not intended to compare the behavior of mosquitoes collected within the same site. Since it is not a comparative study, but a descriptive one, it does not intend to establish statistical differences. All the mosquitoes were captured in the peridomicile of the houses, which includes the stables where the domestic animals stay overnight. After capture, mosquitoes collected from a single site were analyzed as a single universe.
- Line 231 What is the opinion about the human blood collected on those mosquitoes? Were the mosquitoes inside the houses? or the humans were outside? Line 235 in this Mexican work, it seems that the females, which fed on humans, were inside and resting outside. Do you also think is the same pattern?
- Answer. Both hypotheses are possible. However, testing or denying these hypotheses is beyond the objectives of this work. However, the discussion was expanded as follows: “These findings are similar to those previously published in Mexico, where the host selection patterns of An. albimanus collected indoors and outdoors were analysed [29]. Due to the design of our study, it is not possible to demonstrate whether mosquitoes that fed on human blood did so indoors and then went outside to rest or fed on humans outdoors. In this study, human blood was not found in An. crucians, An. neivai or An. punctimacula, species considered as secondary vectors and predominantly zoophilic”.
- Line 239 It doesn't mean that they are not going inside the houses. maybe the smell of insecticide for the mosquitoes still noticeable and they just rather do blood feeding outside. Also we cannot confirm the complete absence of humans during the study period.
- Answer. This is correct. None of those ideas has been suggested in our discussion.
- Line 241 And besides that, the collection frequency and type of trapping.
- Answer. The sentence was modified as follows: “Therefore, a greater affinity for animals as blood source could be explained by the availability of hosts, collection frequency, trapping method and trapping location”
- Line 245 Is this also the case in the study areas of Honduras? in the mentioned worked of Massebo et al., the anophelines were even feeding outdoors, but they were resting indoors.
- Answer. This question is the one that our study tries to help answer throughout the writing.
- Line 247 This mentioned work provides much more information and not only this one. please consider to explore a little more this study.
- Answer. The paragraph has been expanded as follows: “In contrast, several studies have demonstrated that some anophelines select their hosts depending on their availability and not due to strict species tropism. Orsborne et al conducted a systematic review and meta-regression of three major malaria vectors in Africa and showed that HBI was more associated with location of mosquito captures than with mosquito species” [19]
- Line 257 Can this information also be related to the domestication of these species? for instance the Aedes aegypti human blood preference and anthropophilic behavior.
- Answer. This is a possibility, however, the discussion about the coevolution relationships of domestic animals with humans, and the feeding behavior of anophelines is far beyond the objectives of this study.
- Line 260 What could be the reasons for this?
- Answer. There is an open discussion around at least two genetically well-differentiated populations of An. darlingi between Mesoamerica and South America. It is possible that the geographical distance that the Darien plug represents for these two populations has an influence on their behavior. This discussion has been expanded on in our recent paper: (Escobar, D., Ascencio, K., Ortiz, A. et al. Distribution and phylogenetic diversity of Anopheles species in malaria endemic areas of Honduras in an elimination setting. Parasites Vectors 13, 333 (2020). https://doi.org/10.1186/s13071-020-04203-1). This paragraph has been expanded as follows: Differences (genetic and or behavioral) between An. darlingi populations from Mesoamerica and South America could be explained for geographic isolation.
- Line 266 however, with the available references, is this work related to them, or the results also differ?
- Answer. The comparison of our results with the findings of other authors in other countries has been discussed throughout the section. It is indicated that there is much greater availability of food intake data for anophelines in Africa than in the Americas. Of the literature available in America, even more scarce is the one related to the species present in Honduras, such as An. albimanus and An. darlingi.
The following ideas are already contained in the discussion:
1 These findings are similar to those previously published in Mexico, where the host selection patterns of An. albimanus collected indoors and outdoors were analysed [29]
2 Since mosquitoes were caught outdoors at night, the probability of having sourced their blood meal from humans is lower in relation to the number of animals available in the peridomicile. Therefore, a greater affinity for animals as blood source could be explained by the availability of hosts, collection frequency, trapping method and trapping location [32].
3 These data seem to support the premise that blood meal intake reflects host availability rather than host preference [39,40], and provide evidence that these vectors tend to be exophagic, exophylic and zoophagic [41]
4 On the other hand, the results we obtained for An. albimanus are in agreement with previous reports from other Latin American countries, which highlight zoophagic and opportunistic preferences [24,29]
5 Many reports show this behavior of feeding on multiple human hosts [43,44], or different animal species in … Latin America [9,14,16,25,29]
- Line 269 We cannot confirm that in consecutive gonothrophic cycles and availability if mosquitoes would have a different behavior.
- Answer. Unfortunately we do not fully understand the reviewer's suggestion. Could you be more specific please?
- Line 278 This could be seen by different methods, including checking the integrity using an electrophoresis gel.
- Answer. The separation of the DNA by electrophoresis does not allow visualizing the integrity of the DNA from the blood intake of the mosquito, since the DNA of the mosquito is majority and is in good condition after extraction.
- Line 280 however there are appropriate conditions to preserve DNA before extraction
- Answer. Indeed, there are several solutions that preserve the integrity of DNA; however, it was not necessary to use them since there are multiple reports in the literature about it. Furthermore, our results demonstrate that the host DNA was amplifiable. In this paragraph we are offering possible theoretical options to the lack of amplification in some mosquitoes, and in the end, it is indicated that: “A second and more plausible explanation is the existence of other available animal hosts from which mosquitoes obtain their blood meals”.
- Line 281 I may suggest the use of more general primers to amplify, and also sequence the fragment to at least have some information about this unknown host.
- Answer. This is a great contribution from the reviewer. We have considered it for future work, however, the closure of our laboratory due to the covid-19 pandemic prevented us from continuing to deepen these interesting questions. We are sure to address this disturbing question in future work.
- Line 286 Which was not apparently available within the 200m radius, or time of observation to identify them.
- Answer. Wildlife exhibits very cautious behavior towards humans, and therefore it is very difficult to observe them with the naked eye unless special observation techniques are used.
- Line 287 What are the suggestions and alternatives?
- Answer. The following sentence has been added: “Future investigations should consider the use of generic primers from mammals or other groups of animals and subsequent sequencing to discover rare wild hosts.”
- Line 288 According to the provide data: 311 total female captured, 181 were fed, 130 were not, but only 58 were used, what happened to the remaining 72 unfed females?
- Answer. The total number of females apparently unfed (n=130) was not analyzed in this study due to financial limitations. This was not an initial objective of the project and emerged only as a subsequent research question. Therefore, we decided to take only a representative sample to answer the question. We hope we have partially addressed this question.
- Line 300 What would be the difference collecting mosquitoes indoors from those outdoors regarding their infection status. Because the primer used was for all genus and even though no plasmodium (non-human malaric) was found.
- Answer. There is a greater possibility of detecting infections in the vector if mosquitoes are collected inside the houses, because in this way the insects that have a preference for feeding and resting indoors would be included. These mosquitoes were not included in our study due to bias in the capture method. Regarding the second idea, we do not fully understand what the reviewer refers to. We do not understand the relationship between the type of primers and endophilic and endophagic behavior.
- Line 309 I would say that historic data is not showing this, there is a decrease on the number of cases, that can be related to urbanization and better conditions to avoid the contact with the vector/parasite. Having more feeding options and among those not infected with human malaric parasite can also be advantageous, because this mosquito is being able to finish its life cycle without human blood, so this would be breaking the parasite life cycle, interrupting transmission.
- Answer. We thank the reviewer for this new approach to our results. We agree with his/her opinion and have removed this hypothesis from our conclusions. The new paragraph reads as follows: “These results support the hypothesis that malaria vectors in Honduras exhibit opportunistic feeding behavior and underscore the need for additional mosquito-control measures focusing on the peridomestic environment. Such integrated approach would yield more comprehensive data to inform malaria elimination efforts in the country.”

Reviewer 2 Report
Escobar et al. present the results of PCR-based blood meal analyses performed on a small sample of engorged female Anopheles mosquitoes collected in malaria endemic areas of Honduras. This manuscript contributes useful information on the host-use patterns of mosquitoes (mainly out door siting) in Honduras, which had a high malaria epidemic in the past and has managed well to control the malaria in the last few years. The beauty of this manuscript is that (according to the authors claim) this is the first recorded document that is providing the information on blood meal sources of Anopheles species. However, the manuscript suffers from poor significance of the context and methodological limitations. Since, this is long ago established that Anopheles spp. are opportunistic on host feeding, the information generated by this research has low merit in the science.
Below, I am including additional comments for consideration in a revised version of the manuscript.
Background:
L 53,54: I agree that multi-host biting behaviour of Anopheles mosquito helps to maintain their populations but, given the human and some other non-human primates as malaria parasite’s hosts, I doubt that other than these vertebrate hosts helps to maintain the transmission cycles of malaria in that geographical area. Could it be made clearer?
Materials and methods:
Authors have recorded climatic variables (temperature and relative humidity), economic status, but neither the background for recording these info have been stated nor is analysis/interpretation/correlation of these factors with the major results conducted in later sections.
Authors have recorded peri domestic animals within the 200 m radius of the study site. I wonder why the presence of other native animals and birds within the region were neglected, since there is always a chance that these could be suitable blood meal sources for mosquitoes.
I might have missed, but I didn’t see authors reporting the frequency of each animal they surveyed. Without having a denominator, presenting preference (which they are doing in result section) is a technical flaw. Therefore, the results presented here is just a feeding pattern not a preference.
L82: “Each collection……..per site”. The sentence is unclear
L100: “Molecular…describe host seeking preference” …The sentence gives a meaning that molecular tests describe the host seeking preference. I think it would be better to rewords this sentence to state that molecular test was done to identify the blood meal source from five different hosts.
L102: Pizarro et al., have done for guinea pig only
L106: ‘PCR conditions and components’ sounds better than ‘enzymatic reactions’ as table 1 has provided information only on primers, target genome size and annealing temp.
L110 to L119: The method is too detailed. It is worth to present all these details only when the technique is completely different from what other have done. Otherwise, it is better to state in brief and provide a citation of the technique.
No sequencing had been done? How did author exclude false positive amplification?
Results:
As stated earlier, no interpretation of the climatic variables and economy was made. These are not discussed later as well.
Better to present population of human, domestic animals and their ratio. Results say that most frequent blood meal source were from chicken, and cattle, but was it because of their preference or availability is unclear without having the total available population of these hosts.
Since there are universal vertebrate primers that can detect a variety of vertebrate hosts (Flies et al., 2014; Townzen et al., 2008), only five host specific primers were used in this study that had limited the study significantly.
54/181 visibly engorged females blood meal were failed to be amplified, I think this is just a technical flaw of not using universal vertebrate primers.
Discussion:
L236, 237: An. Neivai and An. Punctimacula: small sample sizes that make it difficult to establish clear patterns of host-use or draw substantive conclusions that authors have made.
L276: what is hald?
L278: First reason (DNA degradation) is not applicable for this study. I would better delete that reason and just stay with the other.
Author Response
Reviewer 2
- Background: L 53,54: I agree that multi-host biting behaviour of Anopheles mosquito helps to maintain their populations but, given the human and some other non-human primates as malaria parasite’s hosts, I doubt that other than these vertebrate hosts helps to maintain the transmission cycles of malaria in that geographical area. Could it be made clearer?
- Answer. What we would like to convey is that the availability of hosts (on which anophelines can be fed) could contribute to maintaining vector populations. High populations densities of vectors favor the maintenance of malaria transmission in a given geographic area.
The paragraph has been clarified as follows: “Mixed blood meals from different hosts within a single gonotrophic cycle is also a common behavior for some Anopheles species [17]. This could contribute to maintaining vector populations conducing to high vector densities, which could increase malaria transmission in a given geographic area.”
- Materials and methods: Authors have recorded climatic variables (temperature and relative humidity), economic status, but neither the background for recording these info have been stated nor is analysis/interpretation/correlation of these factors with the major results conducted in later sections.
- Answer. The reviewer is correct. The reason for including this information in the manuscript is that it was pertinent to the description of the anopheline species that were reported in our previous publication (Escobar et al 2020, Parasites & Vectors). Since the data describing some environmental conditions are not directly related to the current objective and are not discussed in this work, they have been removed from both the M&M and results sections. The corrected paragraphs are read as follows:
“Of the six collection sites in the Caribbean region, two were located in Gracias a Dios, a department commonly known as La Mosquitia, which is an ecological region geographically isolated from the rest of the country by the Rio Plátano biosphere. Before the entomological collection, the species of domestic animals present at each site within a radius of 200 m from the collection sites were recorded. Major economic activities at the eight sites were also recorded.”
“Bovines, dogs, pigs and chickens were recorded around households at all collection sites. All departments, except for Gracias a Dios, have significant agricultural and livestock production, with rice, sugar cane and banana as main crops. Subsistence animal husbandry is common, with many families owning chickens and pigs, and cattle to a lesser extent. On the other hand, the main economic activity of Gracias a Dios inhabitants is fishing.”
- Authors have recorded peri domestic animals within the 200 m radius of the study site. I wonder why the presence of other native animals and birds within the region were neglected, since there is always a chance that these could be suitable blood meal sources for mosquitoes.
- Answer. An exhaustive description of all the animals present at the collection sites would have been ideal to deepen the knowledge of the feeding preferences of the anophelines in Honduras. Unfortunately, this approach is highly demanding of time and resources, which we did not have. Furthermore, a complete description of all the fauna near the collection sites could not have been correlated with our results, since we only had specific primers for five hosts (human, and four domestic animals). This work describes the percentage of females that were engorged but that did not amplify for any of the five hosts analyzed, suggesting indirectly that they could have fed on other mammals, birds or even reptiles. Continuing to delve deeper into the role these wild animals play is a goal to be achieved in the near future.
- I might have missed, but I didn’t see authors reporting the frequency of each animal they surveyed. Without having a denominator, presenting preference (which they are doing in result section) is a technical flaw. Therefore, the results presented here is just a feeding pattern not a preference.
- Answer. It is right. The number of animals present at each site was not recorded. Only the presence or absence of animals was recorded, as indicated in section 2.1: “Before the entomological collection, the species of domestic animals present at each site within a radius of 200 m from the collection sites were recorded.”
However, we disagree with the reviewer's point of view for the following reasons:
- a) The definition of preference is: “a greater liking for one alternative over another or others”. The mosquitoes caught outside the houses had various hosts to feed on, and they demonstrated their ability to feed on them without a clear preference for any of them despite the fact that all these domestic animals were located within the same area.
- b) Furthermore, it would be risky to correlate the precise number of animals at each collection site with feeding preferences given the huge differences in body mass, CO2 emission, hair or feather coverage, access to areas of bare skin, etc.
- c) The focus of this study was only to describe for the first time the feeding behavior of anophelines in Honduras. It was not intended to design experiments to demonstrate how a mosquito would behave if exposed to two or more hosts under the same access conditions.
- L82: “Each collection……..per site”. The sentence is unclear
- Answer. The sentence has been modified as follows: “Each mosquito collection was conducted for one night at each site”.
- L100: “Molecular…describe host seeking preference” …The sentence gives a meaning that molecular tests describe the host seeking preference. I think it would be better to rewords this sentence to state that molecular test was done to identify the blood meal source from five different hosts.
- Answer. The sentence has been modified as follows: “Molecular tests were carried out for each mosquito in order to detect five possible food sources”.
- L102: Pizarro et al., have done for guinea pig only
- Answer. We thank the reviewer for noticing this error in the Pizarro reference. The correct quote has been included: Pizarro JC, Stevens L (2008) A New Method for Forensic DNA Analysis of the Blood Meal in Chagas Disease Vectors Demonstrated Using Triatoma infestans from Chuquisaca, Bolivia. PLoS ONE 3(10): e3585. doi:10.1371/journal.pone.0003585
- L106: ‘PCR conditions and components’ sounds better than ‘enzymatic reactions’ as table 1 has provided information only on primers, target genome size and annealing temp.
- Answer. Thanks. Sounds much better. The sentence has been changed as kindly suggested.
- L110 to L119: The method is too detailed. It is worth to present all these details only when the technique is completely different from what other have done. Otherwise, it is better to state in brief and provide a citation of the technique.
- Answer. We thank the reviewer for his/her suggestion; however we believe that detailing the procedure is useful for those who wish to replicate the experiments, especially when the protocols have been adapted and have been slightly modifiedcompared to the originals.
- No sequencing had been done? How did author exclude false positive amplification?
- Answer. The amplification products of each host have been sequenced before in our laboratory, as indicated in the following publication: “Parasit Vectors. 2018; 11: 15. doi: 10.1186/s13071-017-2605-7. Bionomic aspects of Lutzomyia evansi and Lutzomyia longipalpis, proven vectors of Leishmania infantum in an endemic area of non-ulcerative cutaneous leishmaniasis in Honduras”. Positive and negative controls were run for each experiment.
Results:
- As stated earlier, no interpretation of the climatic variables and economy was made. These are not discussed later as well.
- Answer. Thanks. That sentences have been deleted from the document.
- Better to present population of human, domestic animals and their ratio. Results say that most frequent blood meal source were from chicken, and cattle, but was it because of their preference or availability is unclear without having the total available population of these hosts.
- Answer. See point number 4.
- Since there are universal vertebrate primers that can detect a variety of vertebrate hosts (Flies et al., 2014; Townzen et al., 2008), only five host specific primers were used in this study that had limited the study significantly.
- Answer. This is a great contribution from the reviewer. We have considered it for future work, however, the closure of our laboratory due to the covid-19 pandemic prevented us from continuing to deepen these interesting questions. We are sure to address this disturbing question in future work. The following sentence has been added: “Future investigations should consider the use of generic primers from mammals or other groups of animals and subsequent sequencing to discover rare wild hosts.”
- 54/181 visibly engorged females blood meal were failed to be amplified, I think this is just a technical flaw of not using universal vertebrate primers.
- Answer. We agree with the reviewer. That is why this result has been discussed later: “A second and more plausible explanation is the existence of other available animal hosts from which mosquitoes obtain their blood meals. This phenomenon has been widely described both in the Amazon basin (4) and Africa (35,54,55). It is worth noticing that a majority of unidentified blood meal sources was observed in specimens from La Mosquitia (Gracias a Dios), specifically in Anopheles crucians and An. vestitipennis. Since this geographical region is a protected biosphere with little human intervention there exist a highly diverse wild fauna accessible to mosquito bites. Testing for only five blood meal sources is a limitation to our study.”
Discussion:
- L236, 237: An. Neivai and An. Punctimacula: small sample sizes that make it difficult to establish clear patterns of host-use or draw substantive conclusions that authors have made.
- Answer. We agree with the reviewer. That is why a new sentence has been added at the end of the paragraph: “In this study, human blood was not found in An. crucians, An. neivai or An. punctimacula, species considered as secondary vectors and predominantly zoophilic [30,31]. Since mosquitoes were caught outdoors at night, the probability of having sourced their blood meal from humans is lower in relation to the number of animals available in the peridomicile. Therefore, a greater affinity for animals as blood source could be explained by the availability of hosts, collection frequency, trapping method and trapping location [32]. Given the low number of specimens captured from the three species, these hypotheses need to be confirmed later with a larger number.
- L276: what is hald?
- Answer. Typo corrected: “Out of a total of 181 visibly engorged mosquitoes, less than half (n = 54, 29.8%) did not amplify for any of the five animal blood sources analysed.”
- L278: First reason (DNA degradation) is not applicable for this study. I would better delete that reason and just stay with the other.
- Answer. We agree with the reviewer's observation, however we believe that exposing all possible explanations for a phenomenon is an exercise in scientific honesty. Either way, the text explains that this is an unlikely explanation.

Reviewer 3 Report
The manuscript “Blood meal sources of Anopheles spp. in malaria endemic areas of Honduras” by Escobar et al. represents a nice contribution toward understanding the host associations of Anopheles mosquitoes in Honduras, where mosquito faunas are diverse, not well known, and include important vectors of pathogens. This manuscript presents the results of blood meal analysis performed on Anopheles mosquitoes collected at several sites across Honduras. There are several substantial limitations to this study – sampling bias due to mosquito collections at particular resting locations, and the limited range of host species that could be detected by the molecular methods. I was pleased to see that these limitations were well addressed in the discussion, and despite these, the data and results presented in the manuscript are valuable and help to fill existing knowledge gaps in the host use of neotropical mosquitoes. All my comments, below, are minor. Most substantial of these is the need to include the number mosquito blood meals for each species that did not amplify in the blood meal analyses, as this provides additional insight into the potential host use of these mosquitoes since the likely explanation of this is that the host from which these blood meals were derived is not among the species tested for.
Line 48: Indoor not indoors
Line 56: The sentence beginning at Line 56 should be reworded for clarity.
Line 80: More detail is needed on manual aspiration – what type of aspirator was used?
Line 134: Anopheles crucians should be described as Anopheles crucians complex species (see Wilkerson et al. 2004: Ribosomal DNA ITS2 Sequences Differentiate Six Species in the Anopheles crucians Complex (Diptera: Culicidae). I am unfamiliar with the taxonomic details of some of the Anopheles species described in the manuscript, but if any of these represent complexes, this should be updated to reflect that.
Table 2: I suggest including the number of non-reacting/unidentified blood meals for each species as this can also be useful information.
Figure 2: The authors should consider including the proportion of unidentified blood meals in each pie chart. This is useful information, as, for example, blood meals from 16 An. crucians complex mosquitoes were not identified, which suggests they were feeding on hosts other than those tested for. Doing this would provide a more complete picture of the host associations for each species. Also, in the case of An. crucians, the pie chart does not seem to match Table 1, which does not indicate detection of chicken DNA.
Line 213: analysis should be analyses. Were these mosquitoes included in the previous results?
Line 222: vectors should be vector.
Line 231: typo – adequate
Line 276: typo – should this be “less than half”?
Line 288: typo – strength
Italicize species names in citations.
Author Response
Reviewer 3
- There are several substantial limitations to this study – sampling bias due to mosquito collections at particular resting locations, and the limited range of host species that could be detected by the molecular methods. I was pleased to see that these limitations were well addressed in the discussion, and despite these, the data and results presented in the manuscript are valuable and help to fill existing knowledge gaps in the host use of neotropical mosquitoes. All my comments, below, are minor. Most substantial of these is the need to include the number mosquito blood meals for each species that did not amplify in the blood meal analyses, as this provides additional insight into the potential host use of these mosquitoes since the likely explanation of this is that the host from which these blood meals were derived is not among the species tested for.
- Answer. Figure 2 has been updated with the unknown / unidentified blood meals in each pie chart. Table 3 contains the requested information.
- Line 48: Indoor not indoors
- Answer. Done.
- Line 56: The sentence beginning at Line 56 should be reworded for clarity.
- Answer. The sentence has been modified as follows: “Despite their effectiveness, not only these interventions neglect potential vectors that feed and rest outdoors but can also contribute to a selective pressure that reflects a change in the behavior.”
- Line 80: More detail is needed on manual aspiration – what type of aspirator was used?
- Answer. The following sentence has been added: “Mosquitoes at rest on surfaces were aspirated using mouth aspirators with HEPA filter, model 612 (John W. Hock Company).”
- Line 134: Anopheles crucians should be described as Anopheles crucians complex species (see Wilkerson et al. 2004: Ribosomal DNA ITS2 Sequences Differentiate Six Species in the Anopheles crucians Complex (Diptera: Culicidae). I am unfamiliar with the taxonomic details of some of the Anopheles species described in the manuscript, but if any of these represent complexes, this should be updated to reflect that.
- Answer. It is true that An. crucians is currently in the process of taxonomic revision, and the NCBI database classifies it as incertae sedis; however, while clarifying the taxonomy of these insects it is allowed to continue naming it with its original name An. crucians Wiedemann, as demonstrated in more recent literature (10.1093/jmedent/47.4.634 ; https://doi.org/10.1186/s13071-020-04203-1) ; 10.1016/j.actatropica.2018.08.036)
- Table 2: I suggest including the number of non-reacting/unidentified blood meals for each species as this can also be useful information.
- Answer. That information is contained in Table 3 under the “unknown” column. It has also been included in figure 2.
- Figure 2: The authors should consider including the proportion of unidentified blood meals in each pie chart. This is useful information, as, for example, blood meals from 16 An. crucians complex mosquitoes were not identified, which suggests they were feeding on hosts other than those tested for. Doing this would provide a more complete picture of the host associations for each species. Also, in the case of An. crucians, the pie chart does not seem to match Table 1, which does not indicate detection of chicken DNA.
- Answer. Figure 2 has been updated with the unknown / unidentified blood meals in each pie chart. Three pie charts have been corrected also since they had proportion errors. We are thankful to the reviewer for this observation.
- Line 213: analysis should be analyses. Were these mosquitoes included in the previous results?
- Answer. Done.
- Line 222: vectors should be vector.
- Answer. Done.
- Line 231: typo – adequate
- Answer. Done.
- Line 276: typo – should this be “less than half”?
- Answer. Done. Thanks.
- Line 288: typo – strength
- Answer. Done.
- Italicize species names in citations.
- Answer. Done.

Round 2
Reviewer 1 Report
Thank you for revising and consider the suggestions provided. I enjoyed a lot reading the new version and I have no further comments. I only would like to clarify the points that were unclear to the authors as stated below.
As mentioned by the authors on the rebuttal letter, this study aimed to show that the population could be continuously being supported by have more host to blood feed, and due to this constant amount (maybe increasing) number of mosquitoes, this could increase the risk of malaria infection. For this statement, I would expect then to see data as conducted by an entomological surveillance, which keeps track of diversity and abundance of target species over time. This was not the case in this study, because the collections and the observed time had a different purpose. However I understood the attempt to show this possible scenario and this population increase caused by an easier blood meal access to be an indirect way to show it.
Unclear comments:
- Line 269 We cannot confirm that in consecutive gonotrophic cycles and availability if mosquitoes would have a different behaviour.
- Answer. Unfortunately we do not fully understand the reviewer's suggestion. Could you be more specific please?
The original phrase is: “24.9% of 269 mosquitoes were fed on a single host, animal or human”. My point with comment was to mention that not always distinguish whether the mosquito for each gonotrophic cycle would have one host, in different gonotrophic cycles the mosquito may have different hosts and number of hosts per cycle. And in addition, the females that had the unfed appearance could also contribute for a one host/cycle to seems more promiscuous and be grouped with 2 or more hosts/cycle.
- Line 300 What would be the difference collecting mosquitoes indoors from those outdoors regarding their infection status. Because the primer used was for all genus and even though no plasmodium (non-human malaric) was found.
- Answer. There is a greater possibility of detecting infections in the vector if mosquitoes are collected inside the houses, because in this way the insects that have a preference for feeding and resting indoors would be included. These mosquitoes were not included in our study due to bias in the capture method. Regarding the second idea, we do not fully understand what the reviewer refers to. We do not understand the relationship between the type of primers and endophilic and endophagic behavior.
My point with this comment was more related to the possibility to find malaria parasites and because the primer used couldn’t distinguish from human malaria or other type, it would be difficult again to say that mosquitoes using animal blood was also carrying human malaria parasite, in special considering mosquitoes captured indoors/outdoors.
Reviewer 2 Report
The revised manuscript reads much better now. This is an interesting paper coming with newer information on the blood meal pattern of Anopheles spp. in malaria-endemic areas of Honduras.
Authors have shown their disagreement over my suggestion to change the phrase ‘blood-feeding preference’ to ‘blood-feeding pattern’. While I can’t further disagree with the authors decided to use “blood-feeding preference” I have added a note of my understanding of these as below.
“Host-feeding pattern”, interchangeably called host selection, or host-feeding habit denotes a pattern of mosquito feeding on hosts in nature. This is a relative frequency of blood of different types observed in specific bloodmeal samples from a mosquito population within a space and time. Feeding patterns, certainly, are determined by numerous factors as authors have mentioned in their responses, such as including host density, availability, accessibility, irritability, and preference.
“Host preference” is more specifically used to denote the choice of a particular host as a food source, over other species equally and easily available/accessible. This is often studied under laboratory conditions or with animal baits under experimental field conditions.